# TOP-Nav: Legged Navigation Integrating Terrain, Obstacle and Proprioception Estimation

**Junli Ren[1,*], Yikai Liu[1,*], Yingru Dai[1], Junfeng Long[2], Guijin Wang[1,†]**
[1]Tsinghua University ,[2]ShanghaiTech University

**Abstract:** Legged navigation has been widely applied in open-world, off-road, and challenging environments. In these scenarios, estimating external disturbances requires a complex synthesis of multi-modal information. This underlines a major limitation in existing works that primarily focus on avoiding obstacles. In this work, we propose TOP-Nav, a novel legged navigation framework that integrates a comprehensive path planner with Terrain awareness, Obstacle avoidance and close-loop Proprioception. TOP-Nav underscores the synergies between vision and proprioception in both path planning and locomotion control. Within the path planner, we present a terrain estimator that enables the robot to select waypoints on terrains with higher traversability while effectively avoiding obstacles. The locomotion controller tracks the planned waypoints and provides motion evaluations as the proprioception advisor. Based on the closed-loop motion feedback, we offer online corrections for the vision-based terrain and obstacle estimations. Consequently, TOP-Nav achieves open-world navigation that the robot can handle terrains or disturbances beyond the distribution of prior knowledge and overcomes constraints imposed by visual conditions. Building upon extensive experiments conducted in both simulation and real-world environments, TOP-Nav demonstrates superior performance in open-world navigation compared to existing methods. Project page at top-nav-legged.github.io.

**Keywords:** Navigation, Task Planning, Reinforcement Learning

## 1 Introduction

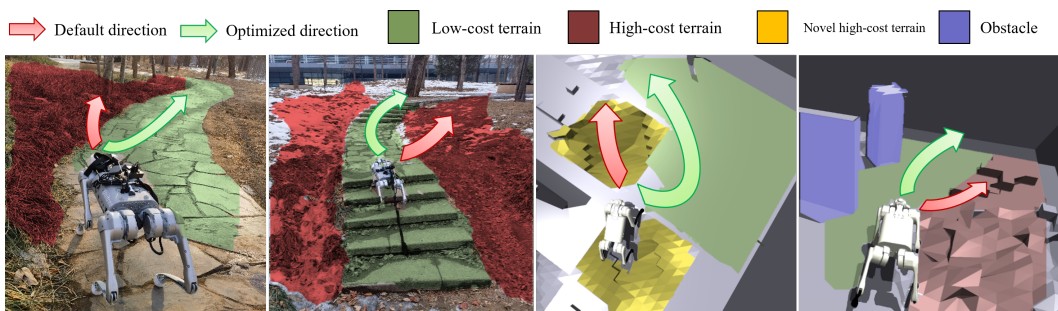

Figure 1: **TOP-Nav** achieves open-world navigation in both simulation and the real world. The robot plans an obstacle-free path on terrains with better traversability. The robot rapidly estimates its traversability for novel terrains based on proprioception experience.

Despite recent advancements in legged locomotion allowing the robots to navigate various terrains based on a simulation-learned strong controller [1, 2, 3, 4, 5, 6, 7, 8, 9], the complexity of currently hard-to-simulate real-world factors makes it impossible for the robot to traverse all potential terrains encountered in reality. As a result, simply connecting the locomotion controller with a vision-based path planner often restricts legged navigation to limited scenarios [10, 11, 12, 13, 14, 15].

8th Conference on Robot Learning (CoRL 2024), Munich, Germany.

An effective solution to overcome these limitations is equipping the robot with terrain awareness: a traversable path can be planned based on the robot's preferences on terrains [16, 17] with an appropriate distribution of contact heights and forces [18]. Unlike obstacle estimation, the distinct terrain features are typically encoded as semantic information, whereas traditional methods collect sufficient data and train segmentation or classification models [19] to learn these features. Nevertheless, compiling an exhaustive catalog of all conceivable terrains and their corresponding walking preferences is impractical [20]. Compounding the issue, the dynamic real-world conditions, such as lighting, humidity, and temperature, may introduce inaccuracies in the correspondence between images and walking preferences, especially when relying exclusively on vision in this context [21].

To address the mentioned challenges of relying solely on vision and ignoring motion states in path planning, we complement the vision-only terrain estimator with online corrections derived from motion evaluations. We construct a proprioception advisor to convey information about the traversability cost of novel terrains and alert the robot to unexpected disturbances, such as invisible obstacles.

By integrating the **T**errain estimator, **O**bstacle estimator, and **P**roprioception advisor, we formulate **TOP-Nav**: a hierarchical path planning and motion control framework navigating a quadruped robot through diverse and challenging terrains proficiently. Within **TOP-Nav**, we develop a terrain estimator trained from previously collected data to inform the robot of terrain traversability. For novel terrains, we compensate proprioceptive history to offer online corrections for the vision-based estimation of unexpected and unknown environmental disturbances. We evaluate **TOP-Nav** both in simulation and on a physical robot, with a comparative analysis against existing legged navigation systems.

## 2 RELATED WORK

### 2.1 Vision and Legged Proprioception Integration

Vision-aided legged navigation has been extensively explored in existing literature [12, 11, 22, 14, 15], with performance heavily dependent on a robust perception module [23]. This reliance makes transferring a specific system to different hardware platforms difficult and costly. Recent research has introduced proprioception to improve task planning, provide comprehensive task observation, and reduce dependence on vision systems. A majority of these works learn proprioception representations along with visual features in simulation and then implement the cross-modal features through end-to-end [24], hybrid [25] or decoupled frameworks [26]. Despite the effectiveness demonstrated in these works, the high-dimensional representation space presents challenges for adaptation to novel scenarios and sim-to-real transfer. Alternatively, Fu et al. [27] introduced a hierarchical navigation framework that derives evaluation scores from motion states, yet overlooks visual observation integration. To mitigate these limitations, we propose a novel approach within **TOP-Nav** by maintaining a series of lightweight cost maps derived from multi-modal observations. This integration achieves a dynamic balance between vision and proprioception. Furthermore, we leverage the learning-based locomotion controller to derive motion evaluations from the value function, offering an efficient solution without additional training.

### 2.2 Terrain Traversability Estimation

Terrain traversability is determined by factors such as terrain geometry, texture, and physical properties [28]. These features could be estimated by identifying the semantic class with a predefined static traversability score [29, 19, 30, 16]. These solutions notably depend on large-scale datasets [31] or are limited to structured environments like urban scenarios [32, 33]. In off-road navigation, the motion states involved in the dynamic interactions between the robot and the environment provide valuable metrics for assessing terrain traversability [34]. These insights have inspired methods that eliminate the need for manual annotation by autonomously deriving terrain traversability from proprioception through self-supervised learning [21, 35, 36, 37, 38, 39]. Nevertheless, the performance of these studies is contingent upon the quality of the collected datasets [28]. Researchers have

proposed various approaches to handling novel observations to emphasize the challenges in unconstrained navigation. For instance, Frey et al. [20] updated the traversability estimation network online with anomalies into consideration. Karnan et al. [40] performs nearest-neighbor search in the proprioception space to align visually novel terrains with existing traversability. Drawing inspiration from those works estimating traversability for novel terrains, we propose a terrain estimator that employs the proprioception advisor as online corrections. Our method diverges from previous approaches primarily in two key aspects: 1) We employ an estimated value function from reinforcement learning to assess terrain traversability, providing a comprehensive evaluation of robot-terrain interactions. 2) We make online corrections on the vision-based estimation without additional training, providing a data-efficient solution for identifying novel terrains.

## 3  Background

Learning-based legged locomotion controllers have been well developed through reinforcement learning, which is generally achieved by updating the policy $\pi_1$ within the asymmetric actor-critic training:

$$\pi_1 \begin{cases} \boldsymbol{a}_t = \pi_{\text{actor}}(\boldsymbol{o}_t^p, \boldsymbol{o}_t^i, \boldsymbol{o}_t^e, \boldsymbol{o}_t^h) \\ c_t = \pi_{\text{critic}}(\boldsymbol{o}_t^p, \boldsymbol{o}_t^i, \boldsymbol{o}_t^e, \boldsymbol{o}_t^h) \end{cases}, \tag{1}$$

the locomotion policy receives privileged observation $\boldsymbol{o}_t^p$, proprioception observation $\boldsymbol{o}_t^i$, scanned dots external observation $\boldsymbol{o}_t^e$ and historical observation $\boldsymbol{o}_t^h$ respectively. $\pi_{\text{actor}}$ is modeled as a Gaussian policy and infers the optimized actions $\boldsymbol{a}_t$ to compute the joint positions $\boldsymbol{q}_{\text{des}}$. $c_t$ stands for the estimated value function from $\pi_{\text{critic}}$, which is updated through:

$$L_t^{\text{critic}} = (c_t - c_t^{\text{targ}})^2 \text{ and } c_t^{\text{targ}} = \sum_{i=t}^{T} \gamma^{i-t} R(s_i), \tag{2}$$

$s_i$ denotes the robot state, $R(s_i)$ represents the rewards accrued at timestep $i$, which commonly includes guiding the robot to track a given velocity command with stable gaits and attitude. Substantial efforts in reward engineering to formulate $R(s_i)$ have enabled the value function $c_t^{\text{targ}}$ to evaluate a comprehensive set of interactions between the robot and its environment. This provides an essential foundation for the proposed proprioception advisor to leverage the estimated $c_t$ for motion evaluations within the path planner.

The complete training paradigm will involve a second stage, training a depth encoder and a student network to reproduce $\boldsymbol{o}_t^p$ and $\boldsymbol{o}_t^e$ from real-world accessible observations $\boldsymbol{I}_d$ (depth image), $\boldsymbol{o}_t^i$ and $\boldsymbol{o}_t^h$. The motion controller will track the target direction $\Delta_{\text{yaw}}$ and velocity command $v_{\text{lin}}$, the control signal is represented by the desired joint position $\boldsymbol{q}_{\text{des}}$.

## 4  Method

### 4.1  System Overview

**TOP-Nav** connects a path planner and a motion controller to tackle the task of legged navigation (Fig. 2). The path planner generates waypoints from an integrated cost map, $\boldsymbol{M}_C$, which is built around the robot and updated through online proprioception and visual observations. $\boldsymbol{M}_C$ offers a comprehensive estimation of external environment that encompasses terrain traversability costs $\boldsymbol{M}_T$, obstacle occupancy $\boldsymbol{M}_O$, proprioception advice $\boldsymbol{M}_P$ and goal approaching $\boldsymbol{M}_G$. The planned waypoint is tracked by the RL-learned locomotion controller. In the following section, we will highlight the terrain estimator and proprioception advisor to demonstrate how the proposed system informs the robot of unexpected obstacles and unknown terrains based on walking experience. Details on the integration of the costmaps within the path planner are provided in the appendix (Section 7.2).

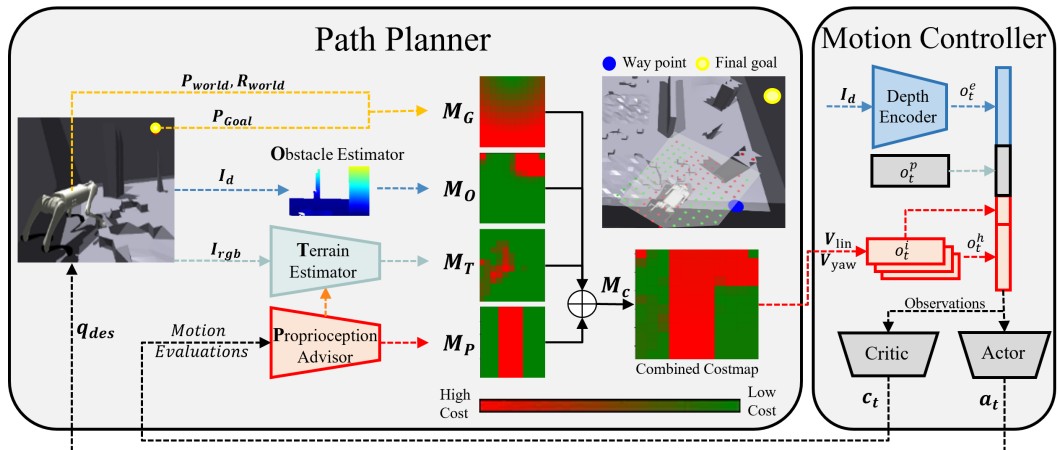

Figure 2: **TOP-Nav** framework: the path planner synthesizes $M_T, M_O, M_P, M_G$ into a combined cost map, from which it computes waypoints based on the overall cost considerations. The controller tracks the desired velocity and provide motion evaluations for the proprioception advisor.

## 4.2  Proprioception Advisor

We design the proprioception advisor to identify motion abnormalities that may arise from unexpected external disturbances. To offer a stable and comprehensive evaluation of the robot-environment interactions, we train the estimated value function (Section 3) to synthesize multiple metrics and incorporate historical observations. As demonstrated in Fig. 3(b), the motion evaluation will decline sharply when the robot encounters a locomotion failure caused by transitioning onto challenging terrain. The reward design and normalization implementation details are provided in Section 7.4.

Within the path planner, we first directly utilize the motion evaluations to enhance the robot's awareness of unforeseen disturbances. This involves estimating invisible collisions ($M_P$) and integrating them into the combined cost map ($M_C$). We propose a continuously varied proprioception cost along the lateral direction (which is the y-axis in $M_P$).

$$M_P(:, y) = \frac{1 - \text{Norm}(c_t)}{e^{k_P(\|y_{\text{base}} - y\|)}}, \tag{3}$$

here $k_P$ is hyperparameter and $c_t$ is the estimated motion evaluation. Since a lower motion evaluation indicates a potentially challenging terrain or an invisible obstacle in the current direction of the robot, we allocate $c_t$ to the centroid column in $M_P$ and decrease it towards the edges, this will prevent the robot from continuing to move forward when its motion is disturbed by an unexpected obstacle or challenging terrain. We further discuss the relationship between the proposed motion evaluations, terrain difficulty, and ground truth motion states in Section 8.3.

## 4.3  Vision-based Terrain Estimation

The proposed terrain estimator first leverages prior knowledge to map the visual observations to the reference terrain traversability. We apply a perspective transformation on $I_{\text{rgb}}$ to map the pixels to bird-eye-views $M_{\text{BEV}}$, which has corresponding coordinates as $M_T$. We then discretize $M_{\text{BEV}}$ into patches (Fig. 3(a)) and assign the same difficulty within each patch. The terrain within the prior knowledge $D_{\text{terrain}}$ is identified through a terrain classification network $\pi_{\text{terrain}}$.

Considering that such identification falls short when faced with unfamiliar terrains, we calculate the predictive entropy to approximate the identification uncertainty [41] and obtain the confidence $Conf$ of current visual estimation accordingly. With $P_i$ denoting the predicted probability of terrain $i$, the predicted explicit $Terrain$ names and confidence of the prediction for the current patch can

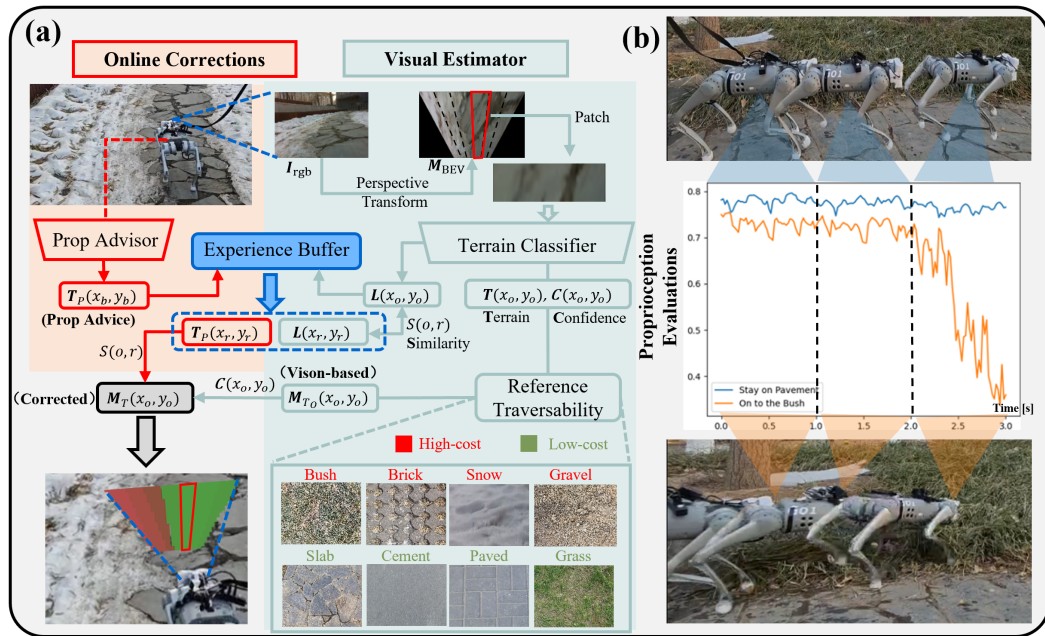

Figure 3: (a) The proposed terrain estimator incorporates a visual estimator and online corrections. (b) The motion evaluations respond rapidly when the robot encounters difficult terrains.

be calculated as:

$$Conf = 1 + \sum_i P_i \log P_i \text{ and } Terrain = \operatorname*{argmax}_i(P_i). \tag{4}$$

then a vision based traversability cost map $\boldsymbol{M}_{T_O}$ could be directly obtained from a pre-defined $(Terrain, Cost)$ mapping. The mapping process leverages experience from previous works [20, 37], where we collect motion evaluations as the robot traverses various terrains in $\boldsymbol{D}_{\text{terrain}}$ (Fig. 3(a)). Terrains with higher motion evaluations are assigned higher traversability scores and lower terrain costs. Detailed discussions are provided in section 7.5.

## 4.4 Online Terrain traversability Corrections

Through the seamless integration of vision and proprioception, we provide online corrections for the vision-based traversability estimation $\boldsymbol{M}_{T_O}$ without additional training (Fig. 3(a)). This enables the robot to terrain awareness beyond the limitations of the collected data $\boldsymbol{D}_{\text{terrain}}$.

We address the mechanism by which the robot recalls the traversability of terrain once it has been seen and traversed within the navigation process. With the robot location $(x_{\text{base}}, y_{\text{base}})$, we record a duration of $1s$ proprioception advice to indicate the terrain traversability cost at the robot location:

$$\boldsymbol{T}_P(x_{\text{base}}, y_{\text{base}}) = 1 - \operatorname{Norm}(\overline{c_{t-k:t}}). \tag{5}$$

Besides $\boldsymbol{T}_P(x_{\text{base}}, y_{\text{base}})$, we can access a latent feature $\boldsymbol{L}(x_{\text{base}}, y_{\text{base}})$ extracted by the classifier $\pi_{\text{terrain}}$ when the same patch was observed a few steps ago, we record both $\boldsymbol{T}_P$ and $\boldsymbol{L}$ at location $(x_{\text{base}}, y_{\text{base}})$ into an experienced list $\boldsymbol{P}_e$. Now given a new observation at location $(x_o, y_o)$, We can find a traversed and seen patch $(x_r, y_r)$ that looks closest to $(x_o, y_o)$ based on cosine similarity:

$$S_{(o,r)} = \operatorname*{argmax}_{r \in \boldsymbol{P}_e} \cos\left(\boldsymbol{L}(x_r, y_r), \boldsymbol{L}(x_o, y_o)\right), \tag{6}$$

Note that we also have the historical traversability cost $\boldsymbol{T}_P(x_r, y_r)$ infered by the proprioception advisor at $(x_r, y_r)$, we can compute a proprioception adapted traversability cost with the similarity

$S_{(o,r)}$ to access the historical proprioception correction $M_{T_P}(x_o, y_o)$:

$$M_{T_P}(x_o, y_o) = \frac{T_P(x_r, y_r)}{1 + e^{-k_{T_3}(S_{(o,r)} - S_0)}}, \tag{7}$$

due to the delay in estimation from the proprioception history, we intend for $M_{T_P}$ to be utilized when encountering novel terrains where $M_{T_O}$ becomes unreliable. Therefore, we normalize the visual uncertainty $U(x_o, y_o)$ into confidence $Conf(x_o, y_o)$ to adjust the contribution of $M_{T_O}$ and $M_{T_P}$:

$$M_T(x_o, y_o) = (\frac{M_{T_O} - M_{T_P}}{1 + e^{-k_{T_4}(Conf - C_0)}} + M_{T_P})(x_o, y_o), \tag{8}$$

here $k_{T_3}$, $S_0$, $k_{T_4}$ and $C_0$ are hyperparameters.

By integrating the adapted $M_T$ as the terrain costmap into the path planner, our system demonstrates rapid adaptability to different terrains and visual conditions.

## 5 Evaluations

### 5.1 Experimental Setup

We assess **TOP-Nav** in challenging navigation environments across both simulations and real-world scenarios, emphasizing the following evaluations: 1) The improvements in navigation performance and locomotion stability achieved by introducing terrain awareness and motion evaluations into the navigation system. 2) The effectiveness of the proposed **P**roprioception advisor. 3) The effectiveness of the proposed **T**errain estimator faced with novel terrains.

**Evaluation Settings** Our simulation experiments are conducted within Nvidia Isaac Gym (Fig. 4). We create a $8 \times 8$ independent navigation cells grid. Each cell is $5m \times 5m$ in size, featuring a robot assigned to a point goal navigation task. The robot and point goal is randomly generated, with a minimum initial distance of $5m$ within the cell. The simulation experiments are conducted 25 times in each of the 64 navigation cells. For real-world evaluations, we conduct outdoor navigation tasks across different scenarios involving challenge obstacles and various terrains, with each scenario replicated 5 times for each method under investigation.

**Metrics:** We evaluate **TOP-Nav** with the following metrics: **SR** (Success Rate): The percentage of successful experiments. We define a success experiment as approaching the point goal into $0.5m$ within 20 seconds; **TD** (Terrain Difficulty): The percentage of average traversed terrain costs, **TD** provides the ground truth traversed terrain costs within each episode; **UT** (Unstable Time): The percentage of unstable motion states ($|roll| > 0.15$ or $|pitch| > 0.15$) in each episode; **VFT** (Velocity Tracking Failure): The average percentage of velocity tracking failures ($\|v_{lin} - v_{act}\| > 0.2$) in each episode; **AEC** (Average Energy Consumption): The average energy consumption ($\tau \dot{q}$) [42] in each successful episode. We calculate the variance across different navigation scenarios to assess the robustness of the proposed method.

**Baselines:** Beyond ablation study, we compare **TOP-Nav** against state-of-the-art legged navigation frameworks and segmentation based terrain estimators. VP-Nav [27] integrates vision and proprioception to develop a collision detector and a fall predictor within the navigation pipeline. GA-Nav [19] achieves terrain segmentation relying solely on vision. Sterling [37] learns terrain traversability in a self-supervised manner by assigning traversability to terrains with similar proprioception representations. However, this approach requires prior training for encountered terrains.

### 5.2 Simulation Evaluations

**Improvements with Terrain Awareness:** **TOP-Nav** empowers the robot to select terrains with higher traversability, leading to a significant improvement in the success rate of navigation: **TOP-Nav** surpasses the VP-Nav baseline by approximately $8\%$ in success rate (*SR*) and achieve a *TD* of nearly $20\%$ (Table 1), which is half of the *TD* achieved in methods without terrain awareness.

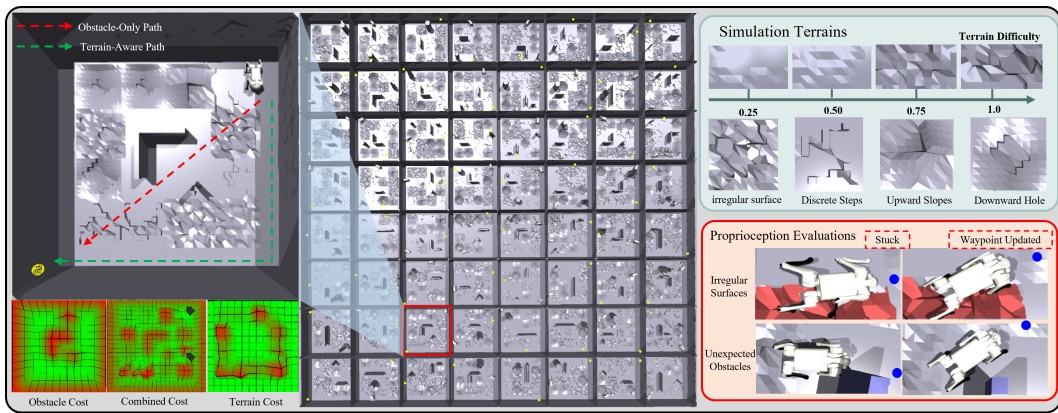

Figure 4: Each navigation cell consists of randomly generated challenging terrains with distinct traverse difficulty, which is marked by the irregularity and complexity of the terrain. The proposed terrain awareness navigation framework plans an optimal path to navigate challenging terrains. The robot demonstrates the capability to recover from unexpected obstacles or irregular terrains with the proprioception advisor.

Table 1: Simulation Results with comparison experiments and ablation study

| Method | SR (%) ↑ | TD(%) ↓ | UT(%) ↓ | VFT(%) ↓ | AEC ↓ |
|---|---|---|---|---|---|
| VP-Nav | $65.62 \pm 2.4$ | $47.18 \pm 0.8$ | $36.18 \pm 0.5$ | $26.09 \pm 0.4$ | $101.80 \pm 49.6$ |
| wo/Terrain | $68.00 \pm 2.6$ | $45.83 \pm 0.7$ | $34.92 \pm 0.6$ | $25.06 \pm 0.5$ | $101.70 \pm 46.6$ |
| wo/Proprioception | $62.75 \pm 1.6$ | $\mathbf{20.47 \pm 0.2}$ | $32.87 \pm 0.5$ | $26.66 \pm 0.6$ | $\mathbf{88.17 \pm 15.3}$ |
| Obstacle-Only | $52.81 \pm 2.3$ | $47.79 \pm 0.8$ | $41.45 \pm 0.8$ | $39.13 \pm 1.0$ | $96.82 \pm 48.0$ |
| **TOP-Nav** | $\mathbf{73.62 \pm 1.2}$ | $22.65 \pm 0.2$ | $\mathbf{27.21 \pm 0.2}$ | $\mathbf{18.14 \pm 0.2}$ | $92.08 \pm 24.4$ |

**Improvements with Proprioception Advisor:** The proposed proprioception advisor could recover the robot from locomotion failures such as getting stuck on irregular terrain surfaces or colliding with unseen obstacles during directional changes (Fig. 4). In contrast to methods without proprioception, our approach exhibits an approximate 11% enhancement in *SR*. Meanwhile, even without terrain awareness, the proposed system outperforms VP-Nav by 2.5%, signifying an improvement in our integration of the proprioception advisor compared to existing methods.

**Advancements in Locomotion: TOP-Nav** achieves the lowest *UT* and *VFT*, indicating that selecting simpler terrains contributes to locomotion stability. We evaluate energy consumption with the assumption that when traversing simpler terrain, the robot should exhibit more natural gaits. As a result, **TOP-Nav** exhibit a 10% reduction in energy consumption compared to methods without terrain awareness.

### 5.3 Real World Evaluations

To evaluate the efficacy of the online corrections for providing the robot with traversability on novel terrains, we conduct experiments using a degraded terrain classifier trained without gravel (Fig 5(A)) and in terrains without any prior information (Fig 5(B)). We demonstrate that the robot can recall previously encountered challenging terrain during the phase of online corrections and plan a waypoint to avoid it. This showcases the fast adaptation ability of **TOP-Nav** to plan an optimal path when faced with novel terrains.

The quantitative results are provided in Table 2. In the *Unkown Gravel* experiments, despite GA-Nav possessing complete prior knowledge and correctly identifying the paved road, our online corrections only lead to a 7% increase in *UT*, mainly due to the forward movement in *phase 1*. VP-Nav allows the robot to exit the gravel terrain with the safety advisor, but it can not retain this infor-

Table 2: Real World Evaluation Results on Novel Terrains

| Scenerios | Unkown Gravel | | | | Slippy Tarpaulin | | |
|---|---|---|---|---|---|---|---|
| | **TOP-Nav** | VP-Nav | GA-Nav | Obstacle-Only | **TOP-Nav** | VP-Nav | Sterling |
| $SR \uparrow$ | **5/5** | 4/5 | 5/5 | 5/5 | **5/5** | 3/5 | 2/5 |
| $UT(\%) \downarrow$ | 14.30 | 26.43 | **7.98** | 42.33 | **1.19** | 2.64 | 3.85 |
| $AEC \downarrow$ | **57.46** | 70.30 | 62.57 | 72.60 | **83.40** | 102.0 | 113.8 |
| $Time(\%) \downarrow$ [1] | **62.72** | 77.90 | 67.42 | 70.64 | 67.62 | **63.05** | 76.52 |

[1] The *Time* metric measures the proportion of time the robot takes to complete the navigation task.

mation. In the *Slippy Tarpaulin* experiments, **TOP-Nav** demonstrates effectiveness in maintaining stability and improving the success rate by avoiding challenging terrain. In the *Slippy Tarpaulin* experiments, **TOP-Nav** demonstarting effectiness in maintaining stability and improving the success rate by avoiding challenging terrain. Among the comparisons, Obstacle-Only keeps moving forward on the slippy surface and exhibit the worst motion performance. We include Sterling in this comparison to demonstrate the importance of online corrections when incorporating proprioception into terrain estimation. Without online corrections, Sterling behaves similarly to Obstacle-Only, unable to adapt to novel terrains and continuing forward. The results against novel terrains highlight the system's ability to perform open-world navigation.

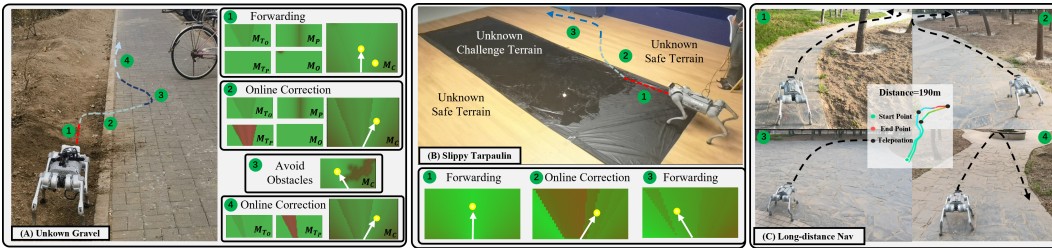

Figure 5: (A) The terrain classifier does not include high-cost gravel in prior. (B) The robot encounters terrains with no prior knowledge, including a slippery detergent surface. We evaluate the effectiveness of the proposed terrain estimator within **TOP-Nav** in the experiments.

In addition, we conducted a consecutive long-distance experiment (Fig. 5-C) in an off-road scenario with grass and gravel on both sides of the path. The robot is initially given a target direction straight ahead, relying on the terrain and obstacle estimator to keep navigating on the paved road. In such scenarios, the challenging terrains help guide the robot in changing its direction, ensuring that it stays on the paved road automatically. More hardware evaluations are provided in Section 8.2 and the supplementary video.

## 6  Conclusion

We present **TOP-Nav**, a legged navigation system that achieves closed-loop integration of visual and proprioception at both task and motion planning levels. Through the extensive quantitative experiments conducted in both simulation and real world, we underscore the success of our system in achieving open-world navigation, surpassing limitations posed by visual conditions or prior knowledge.

**Limitations:** The camera-based vision system in **TOP-Nav** fails in providing robust visual odometry in varying light conditions in outdoor environments. This limitation could be mitigated by integrating LiDAR-odometry into the perception module. Another significant limitation is related to the costmap-based local planner, where the planned waypoint can continuously shift when similar minimum costs arise at different locations on the map. We plan to address this in future work by introducing a diffusion-based task planner, which can integrate different cost modalities while providing a smoother sequence of future waypoints.

**Acknowledgments**

We would like to acknowledge the Visual Computing Lab at Tsinghua University and Unitree Robotics for providing the resources for this paper. We would also like to thank the reviewers for their helpful and insightful comments.

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

# Appendix

## 7 Implementation Details

### 7.1 Task Configuration

To elucidate the task configuration, the robot is given a point goal with its location $\boldsymbol{p}_{\text{goal}}$. The location of the robot is denoted as $(\boldsymbol{p}_{\text{base}}, r_{\text{base}})$. For external observation, we utilize a bio-channel perception module for both the path planner and the controller. This module includes depth ($\boldsymbol{I}_d$) and RGB ($\boldsymbol{I}_{\text{rgb}}$) channels. The path planner computes the desired velocity command ($v_{\text{lin}}, \Delta_{\text{yaw}}$) for the robot, which is tracked by the controller along with the appropraite joint position signal.

### 7.2 Path Planner Details

**TOP-Nav** integrates the combined cost map $\boldsymbol{M}_C$, offering a comprehensive estimation that encompasses terrain traversability cost $\boldsymbol{M}_T$, obstacle cost $\boldsymbol{M}_O$, proprioception advice $\boldsymbol{M}_P$ and the goal approaching cost $\boldsymbol{M}_G$. This integration addresses both visible and unexpected external disturbances against the robot.

To keep a balance between navigation efficiency and safety, we formulate the combination of $\boldsymbol{M}_C$ with dynamic scaling factors as follows:

$$\boldsymbol{M}_C = \boldsymbol{M}_P + \boldsymbol{M}_O + \alpha_T \boldsymbol{M}_T + \alpha_G \boldsymbol{M}_G, \tag{9}$$

$$\alpha_T = \frac{k_{T_1}}{1 + e^{-k_{T_2}(d - d_0)}}, \alpha_G = \frac{k_{G_1}}{1 + e^{-k_{G_2}(t - t_0)}} + k_{G_0}, \tag{10}$$

here $\boldsymbol{M}_i$ are 2-d matrix with a spatial resolution of $0.15m$, $d_0, t_0, k_T, k_G$ are hyperparameters, $t$ denotes current time consuming and $d$ denotes the distance from the point goal at the current step. We accord the highest priority to $\boldsymbol{M}_P$ and $\boldsymbol{M}_O$ since they represent non-traversable locations where the robot cannot pass through. The terrain traversability scale $\alpha_T$ decreases as the robot approaches the target. This design is made considering that, as the robot nears the target ($d$ decreases), taking a detour to avoid a challenging yet traversable terrain would be an inefficient behaviour. Conversely, the goal approaching scale $\alpha_G$ increases with the duration $t$ of the task.

With the integrated map $\boldsymbol{M}_C$, we select the optimal waypoint $\boldsymbol{p}_{\text{way}}$ based on the lowest combined cost.

$$\boldsymbol{p}_{\text{way}} = \underset{\boldsymbol{p} \in \boldsymbol{E}}{\arg\min} \frac{1}{\|\boldsymbol{p}_{\text{base}} - \boldsymbol{p}\|} \sum_{(x,y)=\boldsymbol{p}_{\text{base}}}^{(x,y)=\boldsymbol{p}} \boldsymbol{M}_C(x, y), \tag{11}$$

here $\boldsymbol{E}$ denotes the set of points on the edge of $\boldsymbol{M}_C$. For each point $\boldsymbol{p}$ in $\boldsymbol{E}$, we calculate the average combined cost along the path from the robot location $\boldsymbol{p}_{\text{base}}$ to the edge point $\boldsymbol{p}$. The optimal waypoint $\boldsymbol{p}_{\text{way}}$ is then chosen as the point with the lowest path cost.

After determining the optimized local target $\boldsymbol{p}_{\text{way}}$, the target direction is calculated based on the relative position $\Delta_{\text{yaw}} = \arctan(\boldsymbol{p}_{\text{way}} - \boldsymbol{p}_{\text{base}}) - r_{\text{base}}$, here $r_{\text{base}}$ denotes the yaw direction of the robot base. The linear velocity is constrained to mitigate the impact of high angular variation with $v_{\text{lin}} = v_0 e^{-k\Delta_{\text{yaw}}}$, thereby preventing abrupt and substantial turning at high speeds.

In simulation, $\boldsymbol{M}_C$ is configured with dimensions $(1.65m, 1.5m)$ centered around the robot's location at $\boldsymbol{p}_{\text{base}}^{\text{map}} = (0.45m, 0.75m)$. This design prioritizes a light-weight, real-time updated path planner and maintains consistency with the height map used in locomotion. The environments include randomly generated obstacles and terrains. Each cell is equipped with at least one *wall* obstacle, two *column* obstacles and the remaining space is divided into $1m \times 1m$ sections. For terrain traversability assignments, we uniformly partition the obstacle-free space into difficulty levels [0, 0.25, 0.5, 0.75, 1] and generate terrain with various heights of steps and the intensity of irregular terrain corresponding to the difficulty levels. This difficulty serves as ground-truth walking preferences in simulation.

To obtain a more comprehensive observation in the real world, we configure the cost map $M_C$ with dimensions of $(3m, 3m)$. The location of robot is at $p_{\text{base}}^{\text{map}} = (0m, 1.5m)$ in real world $M_C$. The criteria for success include reaching the goal within the specified time constraints, consistent with the simulation setting.

**Obstacle Estimation and Localization:** In simulation, the robot has access to the ground truth location. For each point $p = (x, y)$, $M_G(x, y) = \|p_{\text{goal}} - p\|$. In real world, we set a target direction $r_{\text{goal}}$ to compute the goal map $M_G(x, y) = \|r_{\text{goal}} - \arctan(p - p_{\text{base}})\|$. We construct the obstacle estimation based on the depth channel $I_d$. The perceived obstacles are converted into point clouds, and for each point in $M_O$, we compute the signed distance $M_{\text{SDF}}$ to the closest obstacles within distance $d_{\text{max}}$, the cost of obstacles therefore can be computed as:

$$M_O(x, y) = \frac{\max(0, d_{\text{max}} - M_{\text{SDF}}(x, y))}{d_{\text{max}}}. \tag{12}$$

For reference, we provide the default values of the hyperparameters used in the path planner in Table 3. Parameters $d_0, t_0$ adjust the weights of $M_T$ and $M_G$ as the robot approaches the target and as time progresses. $k_{G_0}$ keeps a minimum weight for the robot approaching the target. The listed values remain consistent in simulation; however, considering the various conditions in the real world, we make slight changes to these values in different real-world experiments for better validation of the contribution of this work. For instance, in outdoor navigation experiments, we assign a higher value to $k_{T_1}$ to expand the scale of the terrain estimator, which is effective for evaluating the accuracy of the proposed terrain estimator.

The default commanded velcoity $v_0$ is set to $0.5m/s$ in both simulation and the real world. The maximum distance $d_{\text{max}}$ we considered for computing the obstacle map $M_O$ is $0.3m$.

Table 3: Hyperparameters in the path planner

| Parameter | $d_0$ | $t_0$ | $k_{T_1}$ | $k_{T_2}$ | $k_{G_0}$ | $k_{G_1}$ | $k_{G_2}$ |
|---|---|---|---|---|---|---|---|
| Default Value | 0.5 | 0.5 | 1.0 | 2.0 | 0.1 | 0.4 | $-10$ |

## 7.3 Motion Controller Details

The motion controller is implemented following [3, 43]. Both the actor and critic networks within the locomotion policy $\pi$ have hidden layer sizes of [512, 256, 128]. The proprioception observation $o_t^p$ includes angular velocity (3), Orientation (2), velocity commands (3), joint positions (12), joint velocities (12), and the last action (12). We store the last 10 steps of $o_t^p$ into the history observation $o_t^h$. The depth encoder learns the exteroception latent features from onboard observation $I_d$ using a conv-GRU structure. $\pi$ is updated using PPO [44] for 20K iterations, the batch size is 160000 divided into 4 mini-batches. The depth encoder is optimized for 5k iterations. The rewards obtained at each step are the sum of the reward functions listed in Table 4.

## 7.4 Proprioception Advisor Details

The default $\pi_{\text{critic}}$ requires privileged observations, we train another value function estimation network using observable motion states for the proprioception advisor to be deployed on hardware. The estimated motion evaluation $c_t$ is normalized using a sigmoid function. The proprioception advisor is trained with the locomotion policy $\pi$, with an exact same network architechture as $\pi_{\text{critic}}$. The input only includes proprioception observation $o_t^p$ and historical observation $o_t^h$, updated through the MSE loss with $c_t^{\text{targ}}$. The estimated value $c_t$ is normalized through sigmoid function:

$$Norm(c_t) = \frac{1}{1 + e^{2*(2.2 - c_t)}} \tag{13}$$

Table 4: Reward Functions

| | |
|---|---|
| Target Velocity Tracking[1] | $min(\overline{v_{act}}, v_{\text{lin}})$ |
| Target Direction Tracking | $e^{-\|\Delta_{\text{yaw}}\|}$ |
| Orientation Penalty | $-(roll^2 + pitch^2)$ |
| Hip Joint Position Penalty | $-\|q_{hip}\|^2$ |
| z-direction velocity Penalty | $-\|v_z\|^2$ |
| Collision Penalty[2] | $-\sum_{i\in(calf,thigh)}(F_c^i \geq 0.1N)$ |
| Torques Variation Penalty | $-\|\tau_t - \tau_{t-1}\|$ |

[1] $\overline{v_{act}}$ is the projection component of $v_{act}$ in the target direction.
[2] $F_c^i$ is the contact force of calf and thigh indices.

In practice, and we set a minimum threshold $c_{th}$ for the normalized critic value be considered in the path planner, in simulation $c_{th} = 0.8$, while in the real world $c_{th} = 0.5$, $Norm(c_t)$ larger than $c_{th}$ will be set to 1.

$M_P$ is calculated with $k_P = 0.3$, we demonstrate an example of $M_P$ after the robot collides with a wall (with vision and obstacle detection disabled in this trial) in Fig 6. In this scenario, the normalized $c_t$ returns $0.02$, $M_P$ has a width of $0.75m$ along the moving direction of the robot with costs greater than $0.5$.

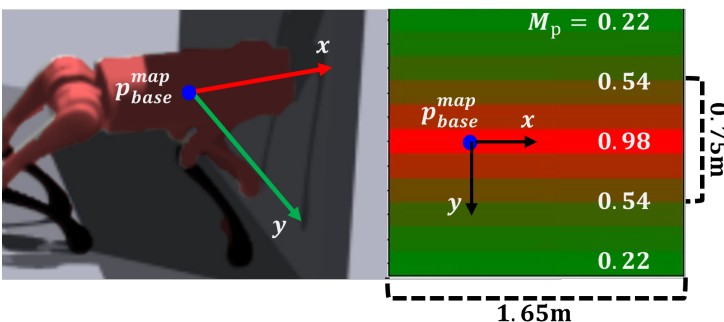

Figure 6: We demonstrate an example of $M_P$ when the robot collides with an obstacle.

Generally, the proprioception advisor engages when locomotion states are compromised. To enable recovery from being obstructed by the unexpected terrains, we have designed the following recovery strategy in simulation: when $Norm(c_t)$ falls below a specific threshold ($0.5$), we initially assign a backward velocity command of $0.5$ m/s for $0.5$ seconds. Next, to address potential terrain-related constraints that may prevent the robot from reaching planned waypoints at its default velocity, we reduce the frequency of the path planner to $0.5$ Hz until the robot reaches the waypoint or makes a directional change of $0.3$ radians.

### 7.5 Terrain Estimator Details

For prior data collection of the terrain estimator, we teleoperate the robot to walk for 5 minutes on each of the concerned terrains. We capture the motion evaluations $c_t$ as well as the first-view observations during these demonstrations. We convert the RGB observations into BEV maps and dividing them into patches. The collected data is labelled based on the corresponding terrain from which it was acquired. The classification network is implemented using the MobileNet backbone and trained on a Nvidia GTX 2080 Ti for 6 hours. The proposed pipeline provides a light-weight inference model of 8 MB to be deployed for onboard computation. During deployment, to approximate the uncertainty $U$ with predictive entropy, We perform $K = 8$ stochastic forward inferences

with different data augmentations as Monte Carlo simulation, We estimate the expectation as the average of the predicted probability $P_i = \sum_k P_i^k / K$.

Based on the collected motion evaluations, we compute the average $c_t$ observed during walks on each terrain. These walking experiences provide reference values for assigning the terrain costs defined by the operator. Table 5 presents the validation accuracy of the terrain classifier, along with the reference terrain cost and the average collected motion evaluations $Norm(c_t)$ for each terrain. Since the terrains encountered in the real world did not exhibit significant differences in motion evaluations, we did not strictly set the reference cost based on linear correlation.

Table 5: Terrain Cost and Classify Accuracy

| Name | Bush | Brick | Snow | Gravel | Slab | Cement | Paved | Grass |
|---|---|---|---|---|---|---|---|---|
| Accuracy (%) | 89.58 | 92.28 | 83.20 | 83.47 | 90.67 | 82.87 | 93.81 | 88.36 |
| $Norm(c_t)$ | 0.481 | 0.523 | 0.445 | 0.507 | 0.581 | 0.580 | 0.580 | 0.572 |
| Cost | 0.7 | 0.6 | 0.9 | 0.7 | 0.0 | 0.0 | 0.0 | 0.2 |

The hyperparameters used for online corrections are fine-tuned based on the distribution of the learned latent features $L$ from the pre-collected data. Figure 7-(b) depicts a t-SNE visualization of the features inferred by the terrain classifier, involving data both within and outside of $D_{\text{terrain}}$. We observe that $\pi_{\text{terrain}}$ can learn distinctive features for specific terrains, showcasing unique clustering in the latent space, even for previously unseen terrain types. The average cosine similarity $S_{(o,r)}$ within each terrain class is 0.95, we set the middle point $S_0$ to be 0.85 and $k_{T_3}$ to be 15. This allows the terrain cost derived from historical experience to decrease rapidly when the similarity falls below 0.8.

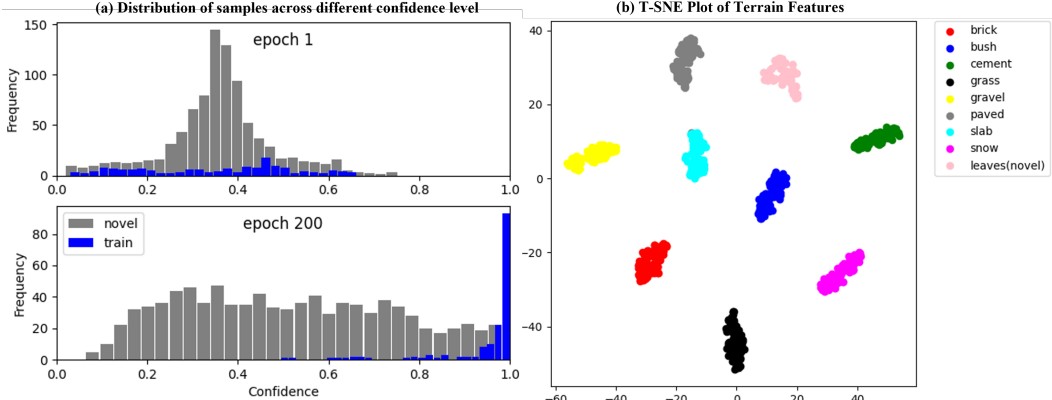

Figure 7: (a) We illustrate the distribution of samples across different confidence levels, with the training of the terrain classifier, the confidence values for known terrains predominantly clustered beyond 0.9, while samples of novel terrains remained dispersed across the confidence axis. (b)T-SNE visualization of the features inferred by the terrain classifier.

To demonstrate the effectiveness of computing confidence in deciding whether the observed terrains is outside of $D_{\text{terrain}}$, we illustrate the distribution of samples across different confidence levels in Fig 7-(a). After 200 steps of training, the confidence values for known terrains predominantly clustered beyond 0.9, while samples of novel terrains remained dispersed across the confidence axis. The confidence is used to adjust the weight of online correct ions in terrain estimation. We set $k_{T_4} = 20$ and $C_0 = 0.9$ in the experiments.

## 7.6 Hardware Details

We implement **TOP-Nav** on the Unitree-Go1 and Unitree-Go2 quadruped robot, with a weight of approximately $12kg$ and dimensions of $645mm \times 280mm$. The robot is equipped with 12 brushless motors, each capable of producing a torque of $35.5Nm$. The perception module is equipped with a RealSense D435i camera, offering simultaneous depth and RGB channels. All computations are processed onboard with an NVIDIA Jetson NX asynchronously, i.e. the locomotion controller operates at a fixed frequency of $50Hz$ and receives the latest depth latent $\ell$ inference from the perception module. The path planner operates at a frequency of $3Hz$.

## 8 Additional Experiments

In this section, we provide additional experiments to justify the effectiveness of the key components in **TOP-Nav**.

### 8.1 Evaluations on Different Evaluation Metrics

This section gives a detailed description of the evaluation choices throughout the simulation and hardware experiments.

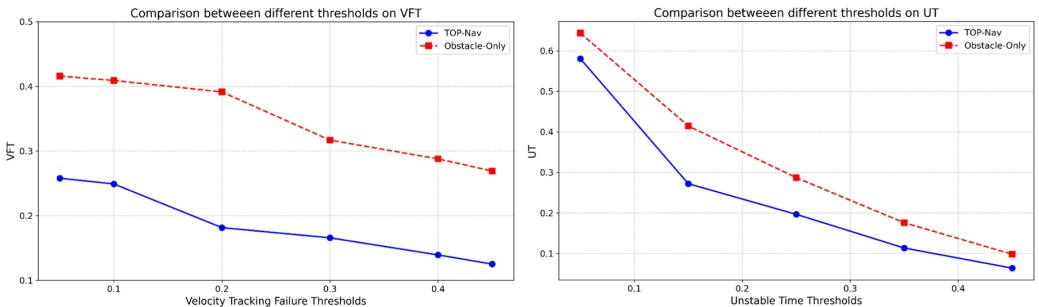

Figure 8: Evaluations on different threholds for **VFT** and **UT** metrics.

We set **The Timeout for Success Rate** at 20 seconds, which corresponds to an expected path length of 10 meters given the commanded velocity of $0.5m/s$. Note that the target goals will be randomly generated at least 5m away from the robot initial location within the 5m x 5m nav-cell. Therefore, 10 meters represents the longest path the robot would plan to complete the navigation episode without backtracking. We set the relatively loose time limit to emphasize the robot's ability to complete the task, as avoiding challenging terrains and obstacles may require taking detours. The experimental results demonstrate that most failure cases occur when the robot gets stuck due to obstacles or challenging terrains. As a result, extending the timeout limit does not lead to a significant change in the success rate. For example, when we raise the timeout limit to 30 seconds, the success rate for **TOP-Nav** increases from $73.62\%$ to $75.93\%$, the success rate for Obstacle-Only raises from $52.81\%$ to $59.39\%$.

We set the metrics of **Unstable Time and Velocity Tracking Failure** to evaluate the advantage in motion stability when the robot follows the planned path using the proposed system. Generally, the robot achieves higher speeds with less roll and pitch when walking on the optimal path. As demonstrated in Fig. 8, the robot performs better on these metrics across different threshold settings.

### 8.2 Evaluations on Vision-Based Terrain Estimator

We conduct additional quantitative hardware experiments on terrains included in the prior datasets (Fig 9), demonstrating superior performance compared to existing legged navigation systems.

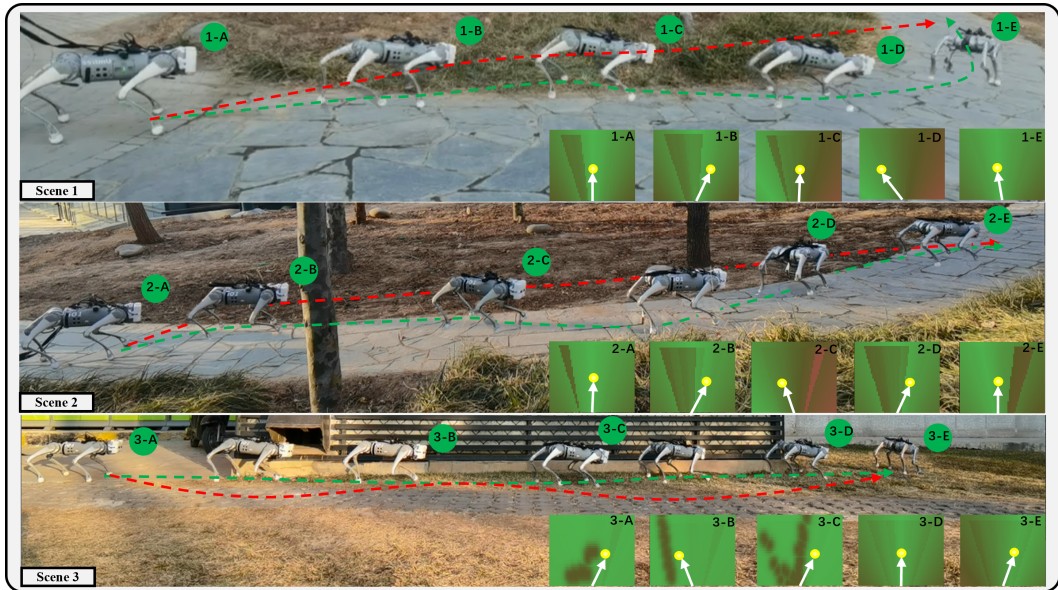

Figure 9: We conduct the evaluations in *scene 1-3*, the green line represents the trajectory of **TOP-Nav**, while the red line represents Obstalce-Only.

The evaluation results are presented in Table 6. **TOP-Nav** successfully completes all 15 trials with superior walking stability and minimal energy consumption. These results affirm the successful deployment of the proposed system on real hardware, allowing the robot to select waypoints on terrains with better traversability and thus execute more natural gaits. In contrast, GA-Nav, trained on RUGD [45], demonstrates notable limitations when confronted with open-world navigation. We observe that the improvements of **TOP-Nav** over GA-Nav mainly stem from: 1) In *scene 1,2*, GA-Nav fails to identify the slabbed road as a low-cost terrain from the bush and gravel. 2) In *scene 3*, GA-Nav incorrectly identifies the lower part of the obstacle as a cement floor, which could be addressed by our integration of obstacles and terrain estimation.

Nevertheless, GA-Nav surpasses VP-Nav and Obstacle-Only, which navigates without terrain awareness. With Obstacle-Only, the robot gets stuck by the stones or Grassroots, leading to a higher time consumption. While VP-Nav effectively alerts the robot to locomotion failures that could be brought by such

Table 6: Evaluation Results on Vision-Based Terrain Estimator

| Method | $SR\uparrow$ | $UT(\%)\downarrow$ | $AEC\downarrow$ | $Time(\%)\downarrow$ |
|---|---|---|---|---|
| GA-Nav | 13/15 | 19.80 | 74.42 | 68.76 |
| VP-Nav | 11/15 | 23.02 | 71.45 | 80.12 |
| Obstacle-Only | 12/15 | 22.75 | 71.09 | 79.72 |
| **TOP-Nav** | **15/15** | **7.79** | **59.07** | **65.03** |

terrains, the absence of terrain awareness prevents the robot from successfully navigating out of these challenging terrains, resulting in inferior locomotion performance.

## 8.3 Evaluations on Proprioception Advisor

In this section, we discuss the relationship between the proposed proprioception advisor and terrain difficulties, along with ground truth reward terms such as velocity tracking and attitude stability.

We conduct a series of straight navigation tasks in simulation, as demonstrated in Fig 10. Each robot's waypoint is set in a straight line, covering terrains of various difficulties $D_l$ and types:

- Irregular Surface: The height of each point in the terrain is randomly changed within the range of $(0.01 + 0.04 * D_l, 0.07 + 0.04 * D_l)m$.

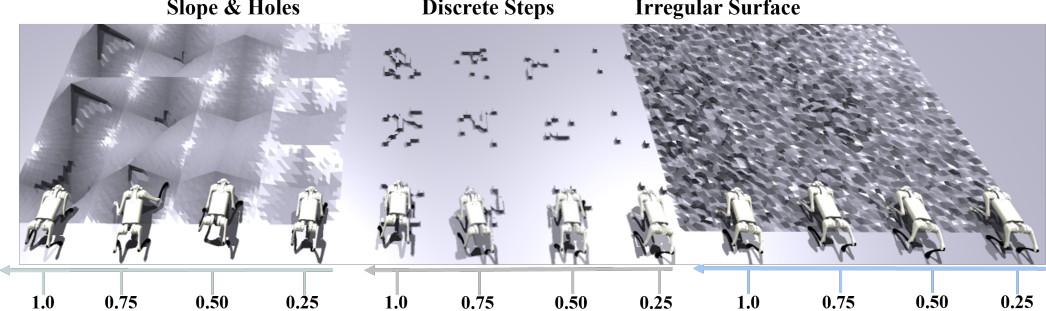

**Slope & Holes**     **Discrete Steps**     **Irregular Surface**

1.0   0.75   0.50   0.25    1.0   0.75   0.50   0.25    1.0   0.75   0.50   0.25

Figure 10: We conduct a series of straight navigation tasks in simulation. Each robot's waypoint is set in a straight line, covering terrains of various difficulties $[0.25, 0.5, 0.75, 1]$ with types covering slopes, discrete steps and irregular surfaces.

- Discrete Steps: This terrain type consists of $12 * D_l$ mall steps within a $1m \times 1m$ area, where each small step has dimensions of $(0.1, 0.1, 0.08)m$.
- Slope & Holes: Randomly switches between upper slopes and holes, where the maximum height/depth of each slope/hole is set as $1.2m * D_l$.

Table 7: Evaluation of the proprioception advisor on different types of terrains.

| Method | Terrain Difficulty | Slope& Holes | Discrete Steps | Irregular Surface | Flat Ground |
|---|---|---|---|---|---|
| Prop Advisor | 0.25 | 0.7560 | 0.7964 | 0.7001 | 0.8600 |
| | 0.50 | 0.3316 | 0.3633 | 0.2589 | |
| | 0.75 | 0.1893 | 0.1614 | 0.1058 | |
| | 1.0 | 0.0573 | 0.0837 | 0.0347 | |
| Decoupled Training | 0.25 | 0.8649 | 0.9836 | 0.9811 | 0.9900 |
| | 0.50 | 0.3076 | 0.7403 | 0.8141 | |
| | 0.75 | 0.2312 | 0.2323 | 0.4831 | |
| | 1.0 | 0.0770 | 0.3230 | 0.2563 | |

We conduct $30k$ steps of navigation for each robot and compute the average motion evaluations estimated by the proprioception advisor. The results are shown in Table 7. We observe that as terrain ruggedness, irregularity, and the presence of obstacles increase, there is a corresponding decrease in the motion evaluations. Although this correlation is not strictly linear, the results provide supportive for us to estimate terrain traversability based on the proprioception advisor.

For comparison, we train an independent motion evaluation function $\pi_{\text{deq}}$, which is decoupled from the training process of the motion controller. $\pi_{\text{deq}}$ is updated using the MSE loss between the ground truth motion states and the network prediction $c_{\text{ref}}$. The ground truth walking states include velocity tracking, orientation penalty, and energy consumption:

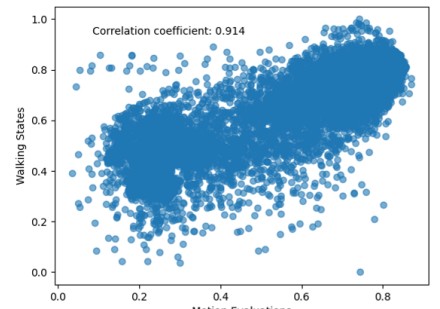

Figure 11: The estimated $c_t$ and the actual motion states of the robot has a strong linear correlation.

$$c_{\text{ref}}^{\text{target}} = min(\overline{v_{act}}, v_{\text{lin}}) - (roll^2 + pitch^2) - 0.001 * \tau \dot{q}, \tag{14}$$

we train $\pi_{\text{deq}}$ for $10k$ iterations, $c_{\text{ref}}^{\text{target}}$ is normalized using a sigmoid function.

As shown in Table 7, $c_{\text{ref}}$ demonstrates a noticeable decline as terrain difficulty increases, yet it does not exhibit a significant advantage compared to the proposed proprioception advisor.

On the other hand, the same terrain may exhibit different traversability for robots with varying locomotion capabilities. Therefore, the evaluation of walking states represents another important metric for assessing terrain traversability. We demonstrate the correlation between the predicted motion evaluations $c_t$ and the ground truth motion states $c_{\text{ref}}^{\text{target}}$ in Fig 11.

The perason correlation coefficient is $0.914$, indicating a strong linear correlation between the estimated $c_t$ and the actual motion states of the robot.

Building upon the discussion above, we validate that constructing the proprioception advisor with critic output enables efficient motion evaluation, including information on both the terrain difficulty and walking states of the robot. Moreover, the proposed advisor can be implemented without the need for additional training or sensors, making it a more appropriate approach to provide motion evaluations for robot path planning compared to existing methods.

## 8.4 Different Velocities

We conducted quantitative experiments in simulation with different velocities $(0.25, 0.5, 0.75, 1.0)$ (m/s), and the results demonstrate that proper velocity design does affects the performance of the navigation system. As demonstrated in Table 8, a slower velocity command $(0.25)$ prevents the robot from reaching the goal on time, resulting in a lower success rate. Moreover, it can be concluded that lower the speed would help decrease the energy consumption.

Table 8: Simulation Results with velocity comparisons

| Velocity (m/s) | $SR$ (%) ↑ | $TD$(%) ↓ | $UT$(%) ↓ | $VFT$(%) ↓ | $AEC$ ↓ |
|---|---|---|---|---|---|
| 0.25 | $49.69 \pm 3.43$ | $26.22 \pm 0.33$ | $26.56 \pm 0.41$ | $25.95 \pm 0.45$ | $65.55 \pm 352.19$ |
| **0.5** | $73.62 \pm 1.21$ | $22.65 \pm 0.19$ | $27.21 \pm 0.24$ | $18.14 \pm 0.16$ | $92.08 \pm 24.39$ |
| 0.75 | $73.91 \pm 2.46$ | $22.22 \pm 0.26$ | $22.53 \pm 0.37$ | $16.79 \pm 0.28$ | $80.32 \pm 46.82$ |
| 1.0 | $72.03 \pm 2.80$ | $21.65 \pm 0.23$ | $41.92 \pm 0.49$ | $21.30 \pm 0.44$ | $123.53 \pm 39.61$ |

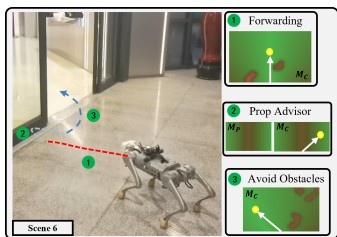

Figure 12: The robot avoids invisble obstacles with the proprioception advisor.

| **Metrics** | VP-Nav | Obstacle-Only | **TOP-Nav** |
|---|---|---|---|
| $SR$ ↑ | 5/5 | 0/5 | 5/5 |
| $UT$(%) ↓ | 1.45 | / | **0.79** |
| $AEC$ ↓ | 42.97 | / | **34.63** |
| $ST$(s) ↓ | 3.30 | $+\infty$ | **1.98** |

Table 9: Evaluation results on the Invisible Obstacles.

On the other hand, setting a too large velocity command will introduce locomotion instability, as indicated by the results with a velocity of $1m/s$. However, conducting a quantitative analysis of the influence of speed is challenging in complex terrains. For instance, in slopes, robots with higher velocities may navigate more effectively, whereas terrains with discrete steps could lead to locomotion failures at faster speeds. Consequently, experiments with velocity commands of $1m/s$ still achieve a high success rate.

The performances vary between $0.5m/s$ and $0.75m/s$ is not significant; therefore, we consider this range to be an appropriate commanded velocity range for deployment. In real-world experiments, we demonstrate the robot's velocity set at $0.75m/s$ in the supplementary video.

## 8.5 Invisible Obstacles

The proposed proprioception advisor provides close-loop feedback to alert the robot to invisible obstacles. We assess this capability in Fig. 12, where a glass wall is located along the planned path. In *phase 1*, the obstacle estimator is oblivious to the presence of the glass wall, and the robot continues moving forward. In *phase 2*, the robot collides with the glass wall, causing a rapid increase in $M_P$ along the direction of movement. As a result, the robot adjusts its waypoint to steer clear of the high-cost area, successfully avoiding the glass wall. The quantitative results are provided in Table 9.

