# OpenReview forum: "TOP-Nav: Legged Navigation Integrating Terrain, Obstacle and Proprioception Estimation"
_robot-learning.org/CoRL/2024/Conference — CoRL 2024_

### Official Review · Reviewer_VJBQ · 2024-07-07
**Los of impressive results but lots of clarification needed.**

**Originality:** 3
**Technical Quality:** 3
**Clarity Of Presentation:** 2
**Potential Impact:** 3
**Recommendation:** 3
**Confidence:** 3

**Review:**

This paper presents a useful idea and show a lot of impressive results. The experiments, especially in the attached video, do a good job of showing the performance of the system. A large amount of integration work is needed to execute this method on a real robot which makes the results very impressive. There is also a good comparison to baselines and sufficient ablation studies to justify design choices. However, I feel that too much is written about the system as a whole when in reality the main contribution is the terrain estimator. As a result, lots of important details about the terrain estimation are missing or confusingly distributed between the main paper and the appendix. For example, I would suggest the authors change Section 3 to discuss less the locomotion policy and more the terrain sensing such as input signals.

Therefore, I mainly have implementation questions and requests for clarification which I will write in the following section. I also recommend reducing content in the paper about the system and instead moving appendix details on the terrain estimator into the main body of the paper.

The main concern I have about the method overall is: how well does the historical terrain experience list scale with large terrain? How often is new terrain information added to the list? What will happen if the robot's position estimate drifts over time?

Finally, there is no limitations section in the paper. My understanding is the paper can be rejected on the grounds of not including a limitations section so I would strongly urge the authors to look at other submissions to CoRL and add a limitations section.

**Quality Of The Limitations Section:**

1

**Questions For Rebuttal:**

1. Add a limitations section.
2. Please address the question of scalability for the terrain experience list and explain how this method works over larger distances.
3. Please clarify the following details:
 - What exactly are the proprioceptive signals used? Joint velocities? IMU? Please explicity describe all sensing including dimensionality.
 - How is the proprioception advisor signal mapped to the Mp like in Figure 10? Why does a collision in the front show low cost behind the wall?

**Robotics Focus:**

4

**Summary Of Paper:**

This paper introduces a framework for estimating traverabilitiy for a legged robot. The method can be updated online with historical experiences.

**Summary Of Recommendation:**

I lean towards accept but there are some issues with the paper which need to be addressed.

---

### Official Review · Reviewer_BuXi · 2024-07-14
**Review of TOP-Nav**

**Originality:** 2
**Technical Quality:** 2
**Clarity Of Presentation:** 1
**Potential Impact:** 2
**Recommendation:** 3
**Confidence:** 4

**Review:**

The proposed method for integrating terrain awareness, obstacle avoidance, and proprioception into reinforcement learning of legged locomotion policies provides an interesting framework for online and continual learning for legged robots. While the authors validate the framework in simulation and in real-world experiments, the thresholds chosen for the evaluation metrics feel arbitrarily chosen. Furthermore, the methodology of the paper is poorly written and hard to understand, making the review of the contribution difficult and the comparison against existing works challenging.

Major Comments:

1. Section 4 is has many grammatical errors and the explanations behind the presented concepts are confusing. For example, statements in Section 4.2 such as "along the y-axis" don't make sense when the coordinate frames have not been introduced, y_base and y in Eq.3 are not introduced, etc. Furthermore, there are mathematical inaccuracies in Section 4. For example, Eq.3 has x as an input, but this does not appear in the evaluation of the right-hand side. Overall I would recommend editing Section 4 so that the presented concepts have a more well-defined motivation and are more clearly explained without requiring the Appendices.

2. The coordinate frames of M are not discussed. Are these cost maps in front of the robot or is the robot centered within them? If they are in front of the robot, how are network predictions for things like the terrain estimator smoothed as the robot moves or as new images are considered?

3. Eq.4 introduces U and T, but both of these values are not used in the paper and only appear in the Appendices.

4. In Section 4.4 it is unclear when the transition between M_To and M_Tp happens. There are other confusing issues in this section as well, such as what the domain and co-domain of the T_P and Norm functions are, why the traversability score T_P is not a binary variable (0 for intraversable, 1 for traversable), or how these local maps M are updated as the robot moves.

5. In Section 5, the thresholds for the evaluation metrics seem to be arbitrarily chosen. For example, why is the timeout for SR (Success Rate) set to 20s? To bolster the authors' claims that their proposed method is in fact better than existing approaches, I would like to see a graph demonstrating the success of all methods as these thresholds are varied (i.e. a graph where the x-axis is timeout in seconds and the y-axis is SR (Success Rate)). The same holds for the UT (Unstable Time) and VFT (Velocity Tracking Failure).

6. The limitations of the proposed methodology are not discussed.

Minor Comments:

 1. In Eq.4 it is incorrect to state U represents uncertainty. Rather, it represents entropy. The authors' may use this as a proxy for uncertainty, but this should be stated clearly.

2. The notation T is overloaded in Eqs. 4 and 5. I would recommend replacing the T in Eq.4 with another variable name.

3. "Traversability" is poorly defined. Intuitively, a region of the robot's environment is either traversable or it is not (1 or 0, respectively). Representing traversability as a continuous variable is thus confusing (i.e. How is a region 0.6 traversable?). Furthermore, this "traversability score" not only varies between robots (i.e. large vs small quadrupeds, bipeds vs quadrupeds), but also has intra-gait variation (i.e. trotting vs static gait). If the RL policy is not constrained to a particular gait, wouldn't traversability vary for the same patch of terrain? For example, if a quadruped tries to run across ice, it would fall (intraversable), but if it walked across slowly it would successfully cross the ice (traversable).

**Quality Of The Limitations Section:**

1

**Questions For Rebuttal:**

1. More clearly explain the methodology in Section 4 and fix grammatical errors throughout the paper.

2. Evaluate the effect of the evaluation metric thresholds on the proposed method and comparisons.

3. Include a more thorough discussion on the limitations of the proposed methodology.

4. Address minor comments.

**Robotics Focus:**

4

**Summary Of Paper:**

This paper presents a method for integrating perception in the form of terrain classification, obstacle identification, and proprioception for the purposes of improving legged locomotion over challenging and potentially unseen terrain.  The proposed framework is validated in simulation and in real-world experiments and is compared against existing approaches. The contributions are further validated via an ablation study examining the contribution of each of the proposed elements.

**Summary Of Recommendation:**

The proposed framework in well validated, however grammatical errors, poorly communicated methodology, and evaluation metric thresholds need to be addressed.

---

### Official Review · Reviewer_CGbG · 2024-07-19
**An interesting “unifying” method for navigation that requires further clarifications**

**Originality:** 3
**Technical Quality:** 3
**Clarity Of Presentation:** 2
**Potential Impact:** 2
**Recommendation:** 3
**Confidence:** 5

**Review:**

**Summary**
I believe the paper is interesting but it’s hard to understand the full method. I get that this is in part because the method builds upon 3 different components, but it’s challenging to grasp the details, as many details are missing in the main text and, if not explained in the appendix, are not explained at all. Lastly, the experiments shown are not particularly challenging for a legged platform to highlight the different modalities supported by the method.

**Strengths**
The overall system proposed is sound. I appreciate that there are experiments performed in closed-loop with the real platform, displaying sensible behavior for a legged platform.

**Weaknesses**
In my opinion, the main weaknesses of the paper are in the design decisions to combine all the sources of information: It is not clear if the goal must be defined within the costmap to be found, it is not clear why the depth information is used twice (in the costmap and the motion planner), it is not clear what terrain estimator is being used, or why defining a virtual barrier in front of the robot for proprioceptive sensing is adequate too. I will provide more details about these questions in the next part.

**Quality Of The Limitations Section:**

1

**Questions For Rebuttal:**

- Please explain why the motion planner needs to be learned if most of the planning happens in the cost map in the end. Why is a traditional local planner insufficient for this purpose?
- Similarly, how does the motion planner leverage depth information twice? (as an input, and as part of the costmap).
- How do you train the method in simulation if the vision input cannot be simulated? Wouldn’t this affect the policies learned by the motion planner?
- The idea of combining different cost maps recalls the work by Wermelinger (https://ieeexplore.ieee.org/document/7759199). Other recent approaches have also been proposed to combine “fields” for navigation (e.g. https://arxiv.org/abs/2201.03938 ). None of them are discussed.
- It is not really clear how the costmap is defined. Is it aligned with the robot’s pose or is it dependent on some fixed frame (like ROS’ grid_map)? How about the magnitude of the cost itself? Is it [0,...,1]?
- How are the semantic classes (from the visual terrain estimator) mapped to costs? Are the values reported in Table 9 hand-crafted?
- The overall presentation of the experiments requires further clarifications. Experiments are presented in an arbitrary way, not really explaining the purpose of them, the conditions, and what’s being assessed.
- Further, without discussing and providing further insights on the failure cases and limitations it’s really hard to assess and understand why the method could overcome challenges of previous approaches. It is not clear to me from the experiments that all the components together ensure the success of the method, since the experiments demonstrate examples that can be solved by one “modality” at a time - the glass door is solved by proprioception and depth sensing (similar to Fu (2022)), the path tracking experiment is similar to Wellhausen (2019, not cited either: https://ieeexplore.ieee.org/document/8627373) or Frey (2023), who solved it with vision only. It would be necessary to provide a further analysis and demonstration on how these different modalities and the overall system design comes together and effectively justify the success of the approach.

Minor text fixes:
- _“Legged navigation is typically examined within open-world, off-road, and challenging environments“_ -> I’m sure that examined is not the right word to use here, but I’m not sure what was the intention of the sentence to suggest an alternative
- _“insimulable”_ -> does not exist, I suggest to write “hard to simulate” instead
- _“open-loop local planner”_ -> a local planner is closed-loop by construction, as it uses online sensing to compute local paths
- _”A comprehensive legged navigation framework Integrating multi-modal”_ -> “integrating” should not be capitalized
- _“As demonstarted in Fig. 3(b)”_ -> should be “demonstrated. There are way too many other typos in the figures and the text, please revise.
- Please do not use red and green together for the figures. They are hard to distinguish for colorblind people.

**Robotics Focus:**

4

**Summary Of Paper:**

The paper represents a method for legged navigation that combines different sources of information, such as RGB vision for semantics, depth estimates for collision avoidance, and proprioceptive sensing for dealing with unobservable obstacles. The approach is based on a RL approach with a costmap representation.  The experimental results are done in simulation and real platforms, showing advantages to other similar methods.

**Summary Of Recommendation:**

I believe the paper needs a general revision: the method is presented in a convoluted manner, important pieces are not explained, and it is not clear if the experiments are really assessing the important aspects and contributions of the work. Apart from presentation issues that were also raised in terms of the wording and typos.

---

### Author Rebuttal · Authors · 2024-08-08

We uploaded a PDF file containing replies to all the reviewers and the revised paper, as well as the Appendix. For the reviewers' convenience, all the comments by the reviewers are written in red, and all the responses are written in black. The revisions are highlighted in the modified manuscript in blue with the "Track Changes" function.

---

### Decision · Program_Chairs · 2024-09-04

**Decision:**

Accept

**Comment:**

In summary, the initial reviews highlight the following strengths and weaknesses of this submission.

Strengths:
- The paper proposes an interesting framework for online adaptation for legged robot navigation.
- The paper provides experimental results with a real platform showing impressive and sensible behavior of the robot.

Weaknesses:
- The paper lacks sufficient details and comprehensibility to fully understand the method.
- The motivation of several design choices is not well explained.
- The experiments are not fully clear and convincing. For instance, it is not clear how the joint consideration of different modalities improves performance, or the thresholds for evaluation metrics seem arbitrarily chosen.
- The paper lacks discussion of limitations, failure cases, and assumptions.

After the author response and discussion phase, the reviewers have been mostly satisfied how the authors adressed their comments, raised their scores and recommend accepting the paper. The authors should additionally consider improving the structure of the paper and providing a definition of traversability as suggested by the reviewers.

The revised paper must not violate the page limit for submission.